# Prevalence and Causes of Anemia in Hospitalized Patients: Impact on Diseases Outcome

**DOI:** 10.3390/jcm9040950

**Published:** 2020-03-30

**Authors:** Maria Luigia Randi, Irene Bertozzi, Claudia Santarossa, Elisabetta Cosi, Fabrizio Lucente, Giulia Bogoni, Giacomo Biagetti, Fabrizio Fabris

**Affiliations:** First Clinical Medicine, Department of Internal Medicine—DIMED, University of Padua, via Giustiniani 2, 35123 Padova, Italy; irene.bertozzi@gmail.com (I.B.); santarossa87@gmail.com (C.S.); elisabetta.cosi@gmail.com (E.C.); fabrizio.lucente@hotmail.it (F.L.); boggg@hotmail.it (G.B.); giacomobiagetti91@gmail.com (G.B.); fabrizio.fabris@unipd.it (F.F.)

**Keywords:** anemia, hospitalized patients, outcome of hospitalization

## Abstract

Anemia is extremely common in hospitalized patients who are old and often with multiple diseases. We evaluated 435 consecutive patients admitted in the internal medicine department of a hub hospital and 191 (43.9%) of them were anemic. Demographic, historic and clinical data, laboratory tests, duration of hospitalization, re-admission at 30 days and death were recorded. Patients were stratified by age (<65, 65–80, >80 years), anemia severity, and etiology of anemia. The causes of anemia were: iron deficiency in 28 patients, vitamin B_12_ and folic acid deficiencies in 6, chronic inflammatory diseases in 80, chronic kidney disease in 15, and multifactorial in 62. The severity of the clinical picture at admission was significantly worse (*p* < 0.001), length of hospitalization was longer (*p* < 0.001) and inversely correlated to the Hb concentration, re-admissions and deaths were more frequent (*p* 0.017) in anemic compared to non-anemic patients. A specific treatment for anemia was used in 99 patients (36.6%) (transfusions, erythropoietin, iron, vitamin B_12_ and/or folic acid). Anemia (and/or its treatment) was red in the discharge letter only 54 patients. Even if anemia is common, in internal medicine departments scarce attention is paid to it, as it is generally considered a “minor” problem, particularly in older patients often affected by multiple pathologies. Our data indicate the need of renewed medical attention to anemia, as it may positively affect the outcome of several concurrent medical conditions and the multidimensional loss of function in older hospitalized patients.

## 1. Introduction

The prevalence of reduced levels of hemoglobin (Hb) in the general population is estimated to be around 30%; the condition equally affects both sexes at all ages, in industrialized as well as in developing countries. In Italy, the estimated prevalence of anemia is 19%, mostly as a mild condition [1,2]. The prevalence of anemia is higher in female subjects, from 15 years of age to adulthood, and in older patients of both genders, affecting 42% of subjects above the age of eighty [3]. 

The most common cause of anemia in the world is iron deficiency (IDA); in contrast, in elderly patients (>65 years of age), multifactorial anemia (chronic kidney disease, nutritional deficiency, occult hemorrhages, gastrointestinal blood loss, use of antithrombotic drugs, ineffective erythropoiesis,) [4] and anemia related to chronic diseases (CDA) have a higher prevalence [2,3], though in some cases the condition may remain unexplained: on a whole, 40% of hospitalized and 47% of institutionalized patients are anemic [5]. Elderly patients with anemia have a reduced physical performance [6], reduced muscle strength leading to a higher incidence of falling [7], to more frequent hospitalizations [8], and to an increased risk of death [9,10,11,12].

Recently, it has been demonstrated that iron metabolism is altered not only in IDA (absolute deficiency) where iron stores are depleted, but also in CDA [13]. During chronic inflammation, hepcidin, a hepatic hormone, increases greatly under interleukins stimuli. Hepcidin inhibits ferroportin activity reducing iron adsorption in the bowel, but also inducing iron sequestration in macrophages and enterocytes [14]. This results in the increase of iron stores (ferritin elevation) and in the likewise reduction of transferrin saturation (TIBC): this condition is named “functional ID” [15].

In recent years, it has been noted that mild anemia in patients with heart failure (HF) is associated with an adverse outcome [16,17,18,19] and that an intravenous iron “one shot” administration improves not only hemoglobin concentration, but also physical mobility, ultimately reducing the number of hospitalizations and mortality [20,21,22]. 

The aims of the present study are: (1) to describe the medical approach to anemia in an internal medicine unit, (2) to correlate hemoglobin concentration to the severity of the underlying disease(s), (3) to assess the potential correlation between low hemoglobin levels and disease outcome, and (4) to propose anemia as an additional indicator to the multidimensional loss of function in older hospitalized patients.

## 2. Material and Methods

This is a monocentric, observational study that evaluates all consecutive adult patients admitted to an internal medicine clinic over a period of 5 months in 2018 (January–May). The Ethical Committee of Padua University City Hospital approved the study (protocol 74480 data 13/12/2019, ex1) and the criteria of Helsinki were followed. 

Two blinded researchers reviewed demographic, historic and clinical data (age at admission, cause of hospital admission, length of recovery, laboratory tests, etiology of anemia, underlying medical treatment, sudden re-admission) of each patient and collected the data in an ad hoc anonymous database. 

At each patient admission, the severity of the disease was assessed with the National Early Warning Score (NEWS 1) [23]: the higher the score, the greater the severity. Length of hospitalization in days and re-admission within 30 days after the first discharge were registered. In June 2019, we checked the on-line registry offices, to know how many of the patients enrolled in the study were still alive. The number of cases in which the diagnosis of anemia and the specific therapy were reported in the discharge letter were also considered. 

Among the 435 patients there were 198 women and 237 men, with a median age of 77 years (range from 19 to 97). Patients were stratified based on age (<65, 65–80, >80 years), on anemia severity in agreement with WHO criteria (mild with Hb <110 g/L, moderate with Hb 80–109 g/L and severe Hb <80 g/L), and on the etiology of anemia.

The following laboratory results were collected: peripheral blood count were available in all 435 cases; serum iron (Fe), ferritin, transferrin saturation (TIBC) in 417 patients; soluble transferrin receptor (sTfR) in 243; reticulocytes count, serum cobalamin (B12) and serum folic acid (fol) in 144 patients. When present, lactic dehydrogenase (LDH), blood in stool, serum erythropoietin (sEPO), pathologic hemoglobin and Coombs test to better explain the cause of anemia were also recorded. 

Iron deficiency anemia (IDA) was diagnosed in the presence of ferritin <30 ug/L and/or TIBC <16% and C-reactive protein (CRP) <5mg/L, and other deficiency anemias with vitamin B_12_ <206 ng/L, and/or folic acid <1.8 mcg/L together or not with ferritin <100 mg/L. The diagnosis of anemia associated to chronic disease (CDA) required the following conditions: CRP >5mg/L, ferritin >100 mcg/L and TIBC <20%, sTfR <1.76 mg/L. Chronic kidney disease (CKD) was considered the cause of anemia when estimated glomerular filtration rate was <30 mL/min. 

Multifactorial anemia was ascribed to patients with more than one of the previous medical conditions.

## 3. Statistical Analysis

The Fisher’s exact test or the X2 test were used to compare categorical variables among the different patient groups. The difference of distribution of continue variables was analyzed using the Mann-Whitney U- test. Correlation was tested with the linear regression. Multivariate analysis was performed with the logistic regression model; for this analysis length of hospitalization was categorized in higher or lower median hospitalization of the whole studied population (8 days). A two-tailed *p* value of <0.05 was considered significant. All statistical calculations were performed using SPSS Statistics 23.

## 4. Results

### 4.1. Total Population Evaluated

The general data of all patients are summarized in Table 1. On average, the prevalence of anemia in our cohort (270 patients, 62%) was similar in both sexes (150 males vs. 120 females) but was higher in the very elderly (>80) compared to patients aging 65–80 years (121 vs. 43). As regards severity, ninety-seven patients (35.8%, 33 females and 64 males, median age 79 y, range 32–97 y) had mild anemia, 133 (49%, 68 females and 65 males, median age 80 y, range 31–96 y) had moderate anemia, and 40 (14.7%, 19 females and 21 males, median age 77 y, range 42–93 y) had severe anemia. In contrast, no significant difference in the prevalence of anemia severity among the three age groups was observed.

The prevalence of NEWS ≥5 at hospital admission was significantly higher (*p* < 0.009) in anemic patients (18.2%) in comparison to non-anemic (9%).

Anemia was statistically more frequent in patients admitted for infectious diseases (66.6% vs. 33.3%) when compared to non-anemic, while the prevalence of anemic patients with cardiovascular diseases did not achieve a significant difference from non-anemic patients (40.6% vs. 59.3%). However, 54 patients with heart failure were anemic, whereas 18 were non-anemic (*p* = 0.0003).

Regardless of the cause of hospital admission, the length of hospitalization was significantly longer (*p* < 0.001) in patients with anemia (10 days, range 1–84), as compared to patients with normal Hb (7 days, range 1–21). The length of hospitalization was inversely correlated to Hb concentration in each patient (Figure 1). Multivariate analysis adjusted for age and NEWS confirmed that, compared to non-anemic patients, all grades of severity of anemia were an independent risk factor for a longer hospitalization (*p* = 0.003, RR = 1.88, CI 95% = 1.3 − 2.85).

Sixty patients with anemia and twenty with normal Hb needed re-admission to hospital within 30 days after discharge (*p* < 0.001) and 25 (9.2%) anemic and 2 (1.2%) non-anemic patients died during hospitalization (*p* < 0.001). At one year after first hospitalization other 78 patients died (56 anemic and 22 non- anemic, *p* = 0.05).

### 4.2. Anemic Patients

Fifty anemic patients (32 mild, 13 moderate and 5 severe—22 females, 28 males) had no sufficient data to clarify the cause of their anemia and, thus, were excluded from this part of the analysis; they represent a prevalence of 18.45% of all the patients with anemia. We also excluded 2 patients with a mild/moderate anemia due to thalassemia, 16 patients with onco-hematologic diseases (5 MDS, 5 AML, 1 MM, 2 NHL, 3 MF) and 11 patients whose anemia was due to hemorrhage, being together 13.1% of our anemic patients.

We considered, therefore, 191 patients (59 mild, 103 moderate and 29 severe—109 males and 82 females) and the causes of their anemia are summarized in Table 2, while Table 3 summarizes the treatment used in different forms of anemia. 

IDA was diagnosed in 28 patients: 6 with mild/moderate anemia received iron supplementation, while all 9 patients with severe anemia received iron and 7 also red cell transfusions. Six patients had a vitamin B_12_ and/or folic acid deficiency, in 2 of them associated with ID; 2 patients received vitamin B_12_ and 2 also iron supplementation. Eighty patients had CDA: 29 patients had moderate, 45 mild and 6 severe anemia; all the 6 patients with severe anemia received a treatment. Within the 15 CKD patients only 2 males received EPO and iron. A multifactorial anemia was diagnosed in 62 patients, however only 15 received anti-anemic treatment. 

An inverse linear correlation between the number of days of hospitalization and Hb level was confirmed also analyzing only anemic patients (*p* = 0.017) and, in multivariate analysis age and NEWS adjusted, low levels of Hb were predictive of longer hospitalization (*p* = 0.013, RR 0.981, CI 95% = 0.966–0.996). 

Treatment for anemia was given to 67 patients (35%): 11 patients (18.6%) with mild, 43 (41.7%) with moderate and 77 (93.1%) with severe anemia. The treatments adopted were in 15.1% of the patients’ blood transfusions, 1% EPO, 4.7% folic acid, and/or vitamin B_12_, 15.7% iron. Within mild anemia patients, none received a blood transfusion or EPO, 8 received iron and 3 vitamin B_12_/folic acid. In moderate anemia, transfusions were used in 7 cases, iron in 28, EPO in 2, vitamin B_12_/folate in 6. Eleven patients with severe anemia were transfused, 12 received iron and 2 had transfusion and iron. 

The presence of reduced hemoglobin and the adopted related therapy was reported in the discharge letter in 33 patients (17.2%): 15 (20.3%) of those admitted for cardiovascular disease (in 7 out of 53 patients—13.2% with heart failure and in 5 out of 18–27.7% with acute coronary disease) 1 with non-specific thorax pain (7.7%) and syncope (5.8%), 8 with infections (12.9%) and 8 with miscellaneous diseases (16.6%).

## 5. Discussion

A large body of studies shows the relevance of anemia, even when mild, in geriatrics and in patients affected by multiple medical conditions [24,25]. In this respect, the decline of hemoglobin levels can no longer be considered the inevitable consequence of aging, as anemia reflects a poor health status and higher morbidity, particularly in the elderly [26].

Our hospital-based study provides further support to these data, as our observations are in line with the percentage of anemia in geriatric patients as reported in other hospital-based studies [27], confirms the high prevalence of anemia in hospitalized patients [28,29,30,31] and its impact on the duration of hospital stay, on the number of re-admissions and, ultimately, on mortality. 

More than three quarters of our patients had a mild/moderate anemia [29]. In these patients, we found anemia in a significantly higher number of males than females [10]. Interestingly, the apparently not-justified higher percentage of males among patients with mild anemia may reflect the normal range of Hb, which is differentiated by gender, while the classification of anemia severity does not consider this difference [1]. To differentiate hemoglobin levels in males and females to establish anemia is a non-sense in patients over 50 years of age, in agreement with Andres et al [26], who considers a cut-off of 120 g/L hemoglobin for anemia in elderly patients, regardless of gender. In our cohort, more than 85% of patients had a moderate or mild anemia [3], the oldest-old patients (>80 years of age) had lower hemoglobin levels than younger patients, but the severity of anemia was similar in all age groups [31].

In agreement with the data of the literature [28,29,32], the length of hospitalization and of a sudden readmission in hospital was less frequent in non-anemic patients than in anemic, and a significant higher percentage of the latter died during or soon after hospital stay. 

Unfortunately, the etiology of anemia was not adequately studied in about 1/5 of our patients, most of them with a mild/moderate (around 95–100 g/L of Hb) form. In general, though these patients were over 80 years of age, their anemic condition was disregarded or not considered, as neither specific treatment nor mention of the condition was made in the discharge letter. However, the importance of anemia in geriatric patients has been recently underlined by the German Geriatric Society [33]. In the remaining patients, a chronic disease was the most common cause of anemia, including chronic kidney disease. However, in none of our patients, serum IL6, that upregulates Hepcidin [34], was tested even if iron deficiency related to Hepcidin increase in “inflammaging” [4] is implicated not only in anemia but also in other age-associated conditions. Many cases of anemia, often severe, were considered multifactorial, while IDA and other were less common. This pattern is similar to the prevalence of different causes of anemia, as reported by Migone et al. [31]. 

Our data show that a commonly adopted treatment for anemia was blood transfusion, regardless of its cause; however, only relatively few patients received a treatment in line with the etiological diagnosis of anemia. In hospitalized patients, cardiovascular diseases are extremely frequent: in our cohort the main cause of hospitalization was heart failure [16,32] and, of note, many of these patients had reduced hemoglobin levels. We surmise that in elderly patients with a cardiovascular pathology the frequent occurrence of multifactorial and sometimes of undefined anemia and the need for a fast correction of anemia suggests the use of blood transfusions also in moderate anemia, as it would result in a rapid, and possibly beneficial, increase of hemoglobin. In our view, ID and CDA should be easily diagnosed and resolved with the administration of rapidly effective iron supplies [35,36,37]. Likewise, vitamin supplementation, namely vitamin B_12_, instead of blood transfusions should be given in patients with vitamin deficiency anemia. A caveat remains for patients with heart failure, as the increase of volemia subsequent to a blood transfusion may cause a worsening of their cardiac function. Furthermore, in infective cases [38] with a high prevalence of anemia, the iron sequestration of bacteria from human tissues is a well-known phenomenon [39,40] to the point that in the emergency settings the severity of anemia has been suggested to differentiate between viral and bacterial infections [41]. Thus, in these situations the treatment of anemia has been questioned, as iron and blood transfusions should facilitate bacterial growth and activity.

It was quite surprising to observe that, within the main diagnosis provided at discharge, in very few cases (around 10%) the presence of anemia, though often mild, was ever acknowledged. Only in patients specifically admitted for severe anemia the problem was adequately addressed. The importance of maintaining normal hemoglobin levels in patients with heart failure has been already underlined [6,20,21,22] but, unfortunately, in our experience, only in a few patients the presence of anemia or iron deficiency was considered and treated.

## 6. Conclusions

This study regards a large cohort of hospitalized patients, providing a significant overview of real-life clinical attention given to anemia. We surmise that the observed frequent disregard of anemia (in particular when mild or moderate), as well the paucity of specific corrective treatment, may be due to the presence of multiple concurring pathologies, particularly in the elderly. In this regard, the treatment of acute relapses of chronic medical conditions, including heart failure, seem the main logical approach to achieve recovery or stabilization, so that the presence of mild/moderate anemia remains a “minor”, if not negligible, problem. However, the restoration and maintenance of adequate hemoglobin levels in these patients may significantly contribute to the clinical improvement of different pathological conditions. A renewed attention to anemia in the elderly and/or in patients with many pathologic conditions, both at the educational and clinical level, would seem justified, as its lack of treatment is associated with a poorer clinical outcome and with the worsening of the multidimensional function in older hospitalized patients.

## Figures and Tables

**Figure 1 jcm-09-00950-f001:**
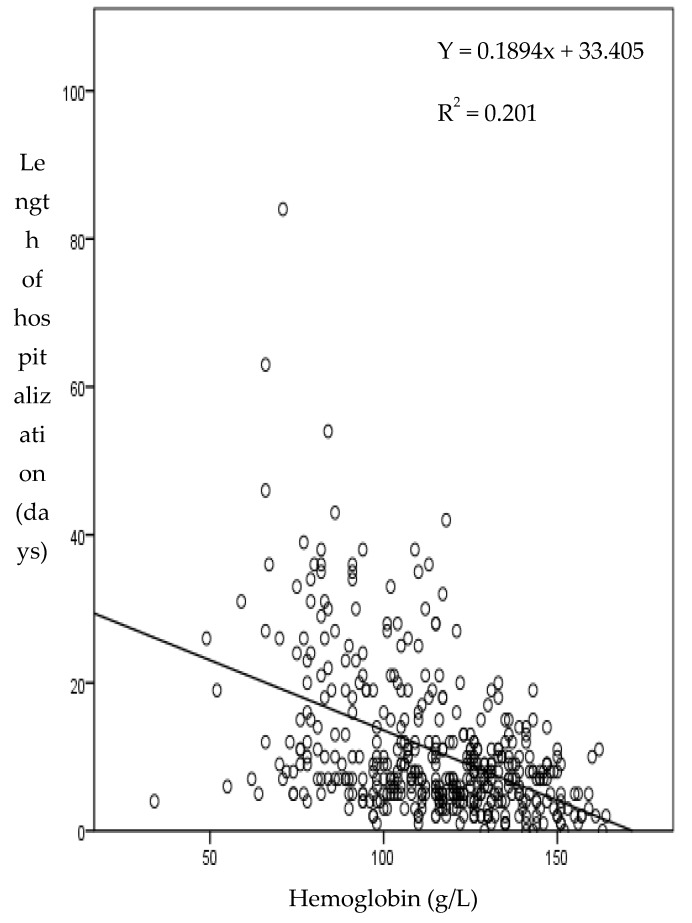
Relation between length of hospitalization and hemoglobin level.

**Table 1 jcm-09-00950-t001:** Baselines characteristics and outcomes of all patients admitted in our Internal Medicine Department over 5 months accordingly with their anemic status.

	Total pts	Non-anemic	Anemic	*p*
Total	435	165 (37.9%)	270 (62%)	
Males/females	237/198	87/78	150/120	
Age (y) median (range) <65 y 65–80 y >80 y	77 (19–97) 95 (21.8%) 178 (40.9%) 162 (37.2%)	72 (19–95) 57 92 66	79 (31–97) 38 86 96	0.001
Median Hb at admission (g/L) (range)	116 (34–164)	136 (120–164)	103 (34–125)	0.001
Severity of anemia Mild Moderate Severe			97 133 40	
patients with NEWS ≥ 5	66 (15.3%)	16 (9%)	51 (18.2%)	0.009
Cause of hospitalization Cardiovascular diseases * Aspecific chest pain Syncope Infections Miscellaneous **	177 55 33 75 68	72 (40.6%) 42 (76.3%) 16 (48.5%) 25 (33.3%) 20 (29.4%)	105 (59.3%) 13 (23.6%) 17 (51.5%) 50 (66.6%) 48 (70.6%)	NS 0.0001 NS 0.0001 NS
Length of hospitalization (day) (range)	8 (1–84)	7 (1–21)	10 (1–84)	0.001
New hospitalization within 30 days (%)	80 (18.4)	20 (12.1)	60 (22.2)	0.001
Deceased during hospitalization (%)	27 (6.2)	2 (1.2)	25 (9.2)	0.017
Deceased within 1 year after discharge	78 (17.9%)	22 (13.3%)	56 (20.7)	0.05

* = acute coronary disease, heart failure, arrhythmia, pulmonary embolism, stroke, other. ** = gastrointestinal, neurologic, renal, hematologic diseases, neoplasms, trauma, chronic obstructive pulmonary disease. Hb = hemoglobin level, NEWS = national Early Warning Score.

**Table 2 jcm-09-00950-t002:** Main general characteristic of our 191 patients stratified by diagnosed causes of anemias.

**Causes of Anemia**	**Pats n. (%)**	**M/F**	**Age y (range)**	**Age Groups (<65/65–80/>80**	**Severity M/MOD/S**
IDA	28 (12.7)	15/13	75 (51–89)	7/13/8	11/8/9
Other deficiency anemias	6 (2)	2/4	84 (74–94)	0/2/4	3/3/0
CDA	80 (36.3)	44/36	79 (31–97)	16/28/36	29/45/6
CKD	15 (6.8)	10/5	81 (48–91)	1/6/8	6/8/1
Multifactorial	62 (28.1)	38/24	80.5 (49–95)	6/25/31	10/39/13

IDA = iron deficiency anemia, CRI = chronic kidney disease, CDA = anemia secondary to chronic disease. M = mild, MOD = moderate, S = severe.

**Table 3 jcm-09-00950-t003:** Anti-anemic treatment adopted in our patients before and during hospitalization.

		**During**				
**Cause of Anemia**	**N° of Pats**	**Red Cell Transfusion**	**Iron**	**Vitamin B12 and/or Folic Acid**	**EPO**	**Total Treated Patients (%)**
IDA	28	9	17	0	0	22 (78.5)
Other deficiency anemias	6	0	2	2	0	4 (66.6)
CDA	80	9	10	5	0	24 (27.5)
CKD	15	1	0	1	0	2 (13.3)
Multifactorial	62	11	1	1	2	15 (24.2)
TOTAL	191	30	30	9	2	

IDA = iron deficiency anemia, CKD = chronic kidney disease, CDA = anemia secondary to chronic disease. EPO = erythropoietin, some patients received more than one therapy.

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
