# Peer review of "Prevalence and Causes of Anemia in Hospitalized Patients: Impact on Diseases Outcome"

_jcm, 2020, doi:10.3390/jcm9040950_

Round 1
Reviewer 1 Report
The manuscript by Randi et al entitled “Prevalence and causes of anemia in hospitalized patients: impact on diseases outcome” urges the need of renewed medical attention to anemia which affects the outcome of several concurrent medical conditions particularly in older hospitalized patients.
The paper is interesting and can be accepted after minor revisions.
- The authors should explain why they did not assay serum IL-6, well known inducer of hepcidin synthesis.
- The authors should cite important papers in which different hepcidin levels has been detected in bacterial and viral infections or in the various phases of infections or in septic inflammation.
- The authors should report in the manuscript the information and registration of the study approved by Ethical Committee of Padua University City Hospital
Author Response
We thank very much the Reviewer for his/her positive evaluation of our work. We appreciated the suggestions made to improve the paper:
Please see lines 213-217 were we have explained why serum IL6 was not evaluated. Please, consider that we have yet cited important papers relating hepcidin and infections: references 40 (ex 38) and 41 (ex 39).
The protocol of the registration of the study has been inserted in the material and method section.
Reviewer 2 Report
This a well done and well written study. The authors were careful to identify shortcomings of this method and were sound in their conclusions and final recommendations.
Although the discussion could be condensed, it would not detract from the manuscript if kept at the current length.
Author Response
We thank very much the Reviewer for his/her favourable evaluation of the paper.
May we note that the positive assessements of the other Reviewers support this positive note
Reviewer 3 Report
Even though the topic (anemia in hospitalized patients) has been often addressed in literature, the authors focus on three very important and interesting points: 1. the association between anemia and the importance of inpatient treatment based on NEWS, 2. the analysis of applied anemia treatment and 3. the reference to anemia at discharge.
recommendations:
- with regard to the older patients, it would be of interest to get information about comprehensive geriatric assessment (CGA)results, espcially in Connection with multidimensional loss of function; if CGA results are not available, it would be interesting to know at least the Barthel Katz Index, which is ususally assessed in all patients at admission, independent of age.
- the importance of anemia in geriatric patients on functionality and clinical outcome has been expressed by the recommendation of the working group anemia of the German Geriatric Society to define anemia a geriatric Syndrome; this article should be included in the text Z Gerontol Geriatr. 2018 Dec;51(8):921-923. doi: 10.1007/s00391-018-1457-x
Author Response
We agree, with the Reviewer, in considering of interest the information about CGA in our patients. However, in our data base it was not available and the Barthel Index was collected only in less than half cases. Therefore, we decided not to consider it: our study is a pointed observation, while the interest should be in evaluating a dynamic Barthel Index, considering that most of our patients are admitted with a low score.
We have cyted the paper of the German Geriatric Society (reference 33)